# What part of the brain is involved in graphic design thinking in landscape architecture?

Yu-Ping Tsai[1], Shih-Han Hung[1☯], Tsung-Ren Huang[2☯], William C. Sullivan[3], Shih-An Tang[4], Chun-Yen Chang[1]*

1 Department of Horticulture and Landscape Architecture, National Taiwan University, Taipei, Taiwan,
2 Department of Psychology, National Taiwan University, Taipei, Taiwan, 3 Department of Landscape Architecture, University of Illinois at Urbana-Champaign, Urbana, Illinois, United States of America,
4 Department of Neurology, Neurological Institute, Taipei Veterans General Hospital, Taipei, Taiwan

☯ These authors contributed equally to this work.
* cycmail@ntu.edu.tw

**Data Availability Statement:** All relevant data is available on Harvard Dataverse Network (https://doi.org/10.7910/DVN/ZHZG5C).

## Abstract

Graphic design thinking is a key skill for landscape architects, but little is known about the links between the design process and brain activity. Based on Goel's frontal lobe lateralization hypothesis (FLLH), we used functional magnetic resonance imaging (fMRI) to scan the brain activity of 24 designers engaging in four design processes—viewing, copy drawing, preliminary ideas, and refinement—during graphic design thinking. The captured scans produced evidence of dramatic differences between brain activity when copying an existing graphic and when engaging in graphic design thinking. The results confirm that designs involving more graphic design thinking exhibit significantly more activity in the left prefrontal cortex. These findings illuminate the design process and suggest the possibility of developing specific activities or exercises to promote graphic design thinking in landscape architecture.

## Introduction

Design thinking is a highly cognitive activity that is widely used in design-related fields to solve the problem in the man-made environment. "Design methodology," as described by Cross [1], is the design process on work and thinking developed by designers through techniques or methods that reflect the knowledge to solve a problem. "Design as a discipline" involves different perceptions in the sciences, humanities, and design thinking, knowing, and practical methods [2, 3]. Hence, we could infer that design thinking involves a series of reasoning mechanisms to select, identify, and then solve the problem [2, 4, 5]. This design process is called "real-world problem solving" with the "select-and-combine model" [6], which relates to the mechanism of conceptual sketches of cognition [7] that might support the frontal lobe lateralization hypothesis (FLLH) in brain activations [4]. Gero and Milovanovic [5] considered that design thinking involves cognition, and researchers have tried to explore the design process through "protocol analysis," "Black Box experiment," and "survey" during design. Goel [4] demonstrated architectural designs that included initial sketching with ambiguous lines,

**Funding:** Funding was received by CYC from the Ministry of Science and Technology, Taiwan [grant number 102-2410-H-002-186-MY3, https://www.most.gov.tw/]. The funder had no role in study design, data collection and analysis, decision to publish, or preparation of the manuscript.

**Competing interests:** The authors have declared that no competing interests exist.

conceptual transformation, analysis, and adjustment of the spatial form, gradually converging the style and precision of its spatial structure to solve current problems. By contrast, Gillieson and Garneau [8] highlighted that graphic design thinking as a design process uses visual communication, such as organization and logic, to define deductive and inductive thinking in space, which also involves personal experiences to recall, reframe, and solve the design problem. Their study addressed the design process by drawing a series of graphics to identify the space in the landscape. Therefore, in this study, we used the term "graphic design thinking" to explore the "Black Box" in landscape architecture design.

Laseau [9] described a thinking process assisted by sketching, while Geol [7] identified four characteristics in design, including problem structuring, preliminary design, refinement, and detail, which use sketching to refine ambiguous ideas in the related arts (e.g., architectural design). This suggests that designers think through sketching to visualize and spatialize a concept map within incomplete ideas and refine those thoughts through abductive thinking [10]. Landscape architecture design consists of a set of design processes, including the scope of the problem, the purpose, and the goal, which interprets "what" and "how" through the selection of material, landscape elements, colors, etc. in the design process to meet needs and create values [11]. The abductive thinking helps designers to develop ideas and connect elements for thought completion [10]. Moreover, the voices of potential users and related stakeholders are important [11]. After the development of the selected solution, the specific idea for actualizing the detailed design will be considered complete. Based on the concept of sketching-assisted thinking proposed by Laseau [9], Geol [7], and Kolko [10], we might infer that, for landscape architects, the landscape architecture design process is associated with the conceptual stages of a project, as landscape architects use sketching to nurture creative ideas, and graphic design thinking refines the ideas to develop and connect landscape elements for completion.

A study proposed a framework of using electroencephalography (EEG), functional magnetic resonance imaging (fMRI), and functional near-infrared spectroscopy (fNIRS) to measure brain activation, which could offer a glimpse of the design process, such as design creativity, design reasoning, and problem-solving, during different design tasks [5]. fMRI may allow a better understanding of design cognition, such as visual and spatial reasoning, in design thinking and creativity [12]. Goel [4] proposed the FLLH, in which the left and right prefrontal cortices (PFC) are responsible for different functions in real-world problem-solving. The hypothesis links the design cognitive process to brain activity in architectural design and planning [13]. The right PFC is involved in planning, visual processing, or reasoning out of the initial concept and preliminary ideas. The right PFC contributes to the abstract, vague, and conceptual aspects of performance. By contrast, after selecting a specific solution and proceeding to further refinement, the left PFC assists in processing specific, clear, and practical information (Fig 1) [4].

The involvement of the PFC in cognitive mechanisms has been proven in neuropsychology [14]. Related research on architecture and interior studies in design results linked with brain activation. The specific design tasks replete with sketching the frame and design items in space in various forms prompted our research design. In their study, Goel and Grafman [13] demonstrated that a patient with frontal lobe lesions who was an architect was unable to perform a preliminary design in sketching the ideas. However, normal subjects could perform the preliminary design, refinement, and detail through the design process in the open-end design. The finding highlights the role of the right PFC in design thinking tasks, consistent with other findings on interior design [12, 15, 16]. Alexiou et al. [12, 15] showed that when subjects in their study used a trackball mouse to move objects during interior design tasks (with no restriction), certain brain areas (e.g., dorsolateral prefrontal cortex (DLPFC), anterior cingulate cortex (ACC), middle frontal gyrus, and middle temporal gyrus) were more active than

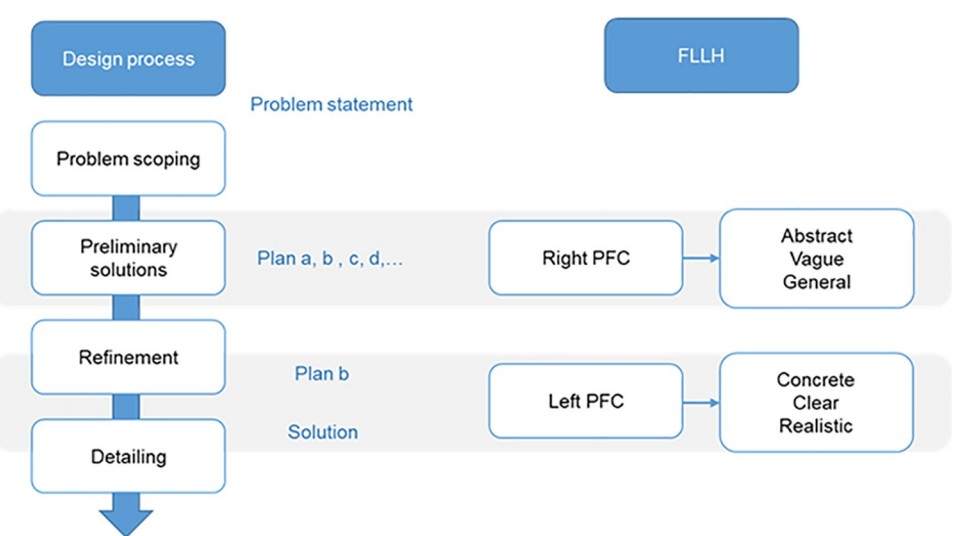

**Fig 1. The design process and corresponding brain reactions (modified from Goel, 2014).**

during problem-solving tasks (with the conditions). These results were consistent with those of Gilbert et al. [16], who found that the right DLPFC and left frontal lobe were more active, stimulating potential solutions within visual imagery in design tasks. Further, these studies indicated that during design tasks, the activation of ACC could relate to cognitive and emotional functions [12, 15]. The DLPFC and ACC are not only involved in executive functions, such as visual imagery and semantic processing, but also work together to construct new ideas that respond to problem solving in design tasks [12, 15].

During design, designers use sketching to draw preliminary ideas and refine them. Landscape architecture design deals with landscape elements, such as water, plants, and pavements, to integrate elements into a whole that form the compatibility in the space. Architectural design involves structures and coordination in buildings. The design process might be similar. A landscape architecture design involves problem structuring, preliminary design, refinement, and detail, which, as proposed by Geol [7], could represent the graphic design thinking process of design with different elements in the landscape. Using fMRI could shed light on the "Black Box" in the brain activation of landscape architects during the design process. To test the association between the different design processes and brain lateralization, this study explored the neural mechanisms associated with the graphic design thinking process by using a fMRI scan to identify parts of the brain that predominate during this phase of landscape architecture design. According to Goel [4], the preliminary solution (or idea generation) phase activates the right PFC, whereas the refinement and detailing (or idea production) phase activates the left PFC. Thus, we hypothesized that the PFC would be activated during the landscape architecture design process as a key brain region that controls the design process.

## Materials and methods

### Participants

The participants (N = 24; 10 males; mean age = 34.50 years; SD = 2.03 years) all had at least three years of training in landscape architecture design. All were right-handed, with normal vision and hearing; none had a history of neurological disorders or cardiovascular disease, and all were screened for MRI compatibility. Each participant gave their written informed consent to the protocol #201411HM024 "*Neural Correlates of Landscape Design Creativity*: *An fMRI*

*study*" as approved by the Research Ethics Committee, Division of Research Ethics, Office of Research and Development, National Taiwan University.

## Apparatus

To test brain activity during landscape graphic design thinking, respondents were placed in an fMRI machine and asked to draw. However, as it is difficult to draw while lying in the fMRI machine, most researchers in previous studies asked participants to verbally describe, silently imagine their ideas, or click and remove the design items during the experiment, later drawing or writing down their ideas from memory outside the machine [12, 15–17]. In the present study, as the time difference between brain activity and reporting was problematic, we created a tilting acrylic table in the fMRI machine using a shoulder fixation cushion to help participants draw their designs while their brains were being scanned. The table was positioned over the participant's waist (Fig 2). The experimental tasks on the table were printed and bound on paper and were easily visible with a double mirror on an overhead coil. Participants turned the page to the next task after hearing a beep in their headphones. While performing the tasks, the participants were asked to move only their lower arm to reduce movement and unnecessary scanning noise.

## Stimuli

The tasks involved two kinds of stimuli. The first task was to complete landscape plans from the book *From Concept to Form in Landscape Design* [18]. The second task involved illustrations containing five geometric shapes (Fig 3) to form a landscape, which included in holistic work of visual imagery, abstract reasoning, or refinement in the graphic design thinking procedure and complete in the design task. A total of four "viewing landscapes" and four "landscape architecture designs with geometric shapes" were arranged differently for the subjects in the research design.

## Procedure

The experiment was designed to understand brain activity during the landscape architecture design process. The landscape graphic design process includes preliminary ideas, in which the

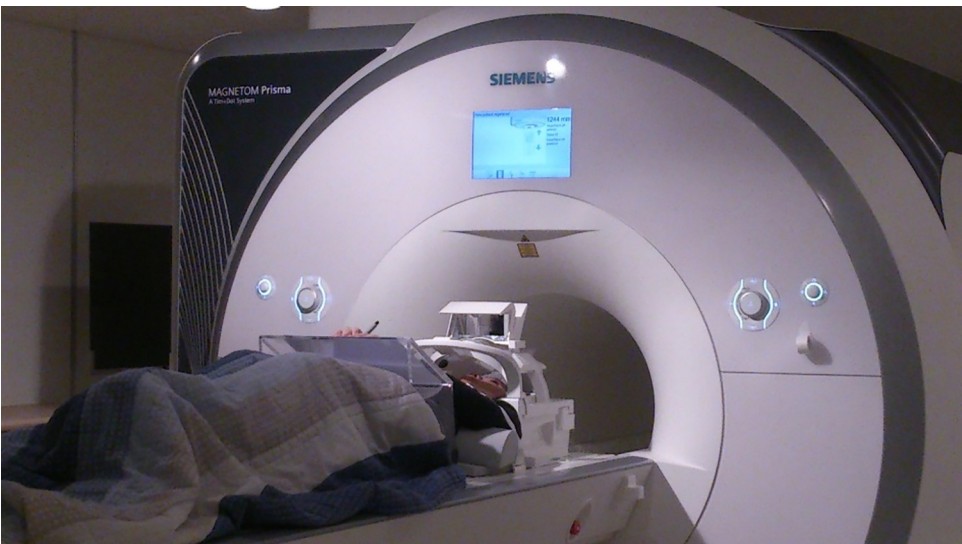

**Fig 2. The scanning environment.**

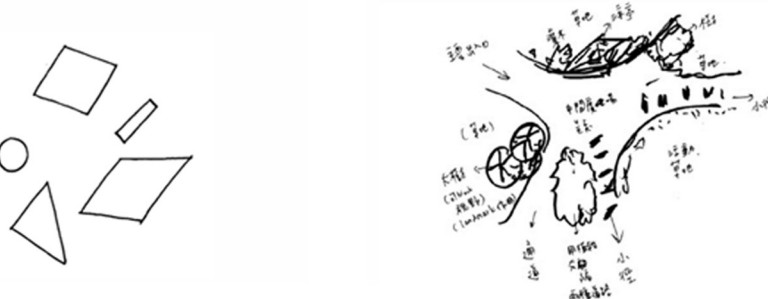

**Fig 3. Examples of geometric cues as experimental stimuli.**

participants think of initial solutions to the refinement problem-solving step. In the refinement phase, the subjects drew and refined their ideas during graphic design thinking, which we assumed activated the left PFC. Two types of tasks were provided in the experiments. The first task involved drawing without graphic design thinking (Task A-control), while the second task involved design cued by simple geometric forms for designing a landscape (Task B-design). To manage the effects of motion, each stimulus was presented twice, yielding four experimental conditions: "viewing" a complete landscape plan (A-1); "copying" this landscape plan on tracing paper (A-2); "preliminary ideas," using geometric illustrations to envision a landscape architecture design (idea generation) (B-1); and "refinement" of the ideas, which uses designing skill to produce the preliminary ideas, reasoning, and drawing of landscape elements (idea production), consistent with the description of graphic design thinking assisted by sketching (B-2). The participants were requested to draw by their hands only during the experiment in the "copy drawing" and "refinement" conditions.

The research design followed the characteristic of design [7]. In the "viewing" condition, participants were asked to imagine that they were in the environment that the plan presented. In the "copy drawing" condition, participants were asked to trace the landscape plan on a tracing paper without any design. In the "preliminary ideas" condition, participants were asked to use several geometric illustrations to envision a landscape architecture design in their mind. In the "refinement" condition, participants were asked to draw the design they thought of in the preliminary ideas session. Based on the different stimuli, participants first copied the landscape plans and then used the geometric forms to develop a plan, section, or perspective drawing of their design according to their own preference. Each condition allowed 60 seconds for completion. During fMRI scanning, the participants completed two runs, which meant that each functional scan included four sessions and lasted for 558 seconds. The participants were randomly assigned to one of the counterbalancing task sequences, ABAB BABA or BABA ABAB (Fig 4).

## Evaluation of the "refinement" stage in graphic design thinking scores

To validate the results of the brain activation experiment, behavioral performance was measured to verify the level of "refinement" in graphic design thinking. Since graphic design thinking assisted by sketching to form design ideas, which might relate to developing creative design ideas, we used the concept of creativity to evaluate it. Previous research has indicated that qualitative research on creativity should be conducted in domain-specific ways [19]. Because there is no standard method for evaluating creative landscape architecture design, we used the indicators in the Abbreviated Torrance Test for Adults (ATTA) as our evaluation instrument. The ATTA is a validated creativity assessment instrument that asks subjects about

**Fig 4. The fMRI experimental process.**

the suitability of writing or drawing in a test of graphic design thinking. The content and volume of ideas in the drawings are used as objective indicators of a designer's creativity. According to Guilford's concept of creative thinking, four indicators—fluency, originality, elaboration, and flexibility—were used to rate the volume of ideas, novelty of concepts, level of detail, and diversity of ideas.

To evaluate the level of refinement in participants' graphic design creative thinking, we invited 10 senior landscape experts to assess the drawings on a 10-point Likert scale. Based on Guilford's concept of creative thinking and the Torrance test, creativity levels were assessed in terms of three components: fluency, originality, and elaboration [20]. *Fluency* refers to the quantity of ideas produced; creative people are able to produce more ideas. *Originality* refers to ideas that are unusual, novel, unique, and different from others. *Elaboration* refers to the details considered and depicted over and above the core concept. Flexibility, the fourth component of creativity in the Torrance test, which tests the diversity of ideas, was not included in this study, as the experimental tasks provided geometric forms that limited the respondent's flexibility. The score of each participant's "refinement" level in the graphic design thinking was based on the average of the ten experts' scores on the three relevant components.

## fMRI data acquisition

Images were acquired by a 3T SIEMENS MAGNETOM Prisma MRI with a 20-channel head coil. For each participant, a T2-weighted anatomical was obtained (TR = 9530 ms, TE = 103 ms, flip angle = 150˚, field of view = 192×192 mm$^2$, x-y voxel size = 0.66×0.5 mm$^2$, 3mm thick). Two-dimensional echo-planar images (EPI) were acquired with a GRAPPA acceleration factor of two at repetition time TR = 3000 ms, echo time TE = 30 ms, flip angle = 90˚, field of view = 192×192 mm$^2$, matrix size = 64×64×45, and an effective resolution of 3×3×3 mm$^3$. In total, 45 EPI slices were sampled in a bottom-up, interleaved order.

## fMRI data analysis

FMRI images were preprocessed and analyzed using SPM8 software in MATLAB. The data for each participant were preprocessed as follows. First, slice timing was used to correct the timing of the functional series by using the middle slice as the reference point. Second, images were realigned to the first scan to correct translational and rotational motion within the subject throughout the time series. Third, using coregistration algorithms, the anatomical image

(higher resolution image) was coregistered to functional images, providing better normalization to the Montreal Neurological Institute (MNI) template. As the next step in normalization of the functional image, the segment procedure divided the anatomical image into gray matter, white matter, and cerebrospinal fluid. Next, the normalization step transformed the realigned functional image data from each individual subject to fit a standardized space, enabling the comparison of brains of varying shapes and sizes. Lastly, the smoothing procedure compensated for the remaining difference between subjects by applying an $8\times8\times8$ mm$^3$ Gaussian smoothing kernel filter.

Whole-brain analyses were estimated voxel by voxel according to a general linear model. Individual statistical maps modeled the time series using regressors and covariates. The covariates were yield movement parameters during the realigning process to control variance due to head movement. The regressors of interest included viewing (A1), copy drawing (A2), preliminary ideas (B1), and refinement (B2) sessions for each participant. In a second-level analysis, the group random effect was assessed for each contrast between participants. Significant regions of brain activation were evaluated by a one-sample t-test to determine whether the mean activation value across participants differed significantly from zero.

According to the assumption of pure insertion, neural structures underlay a single process. First, we compared the difference between "preliminary ideas (B1)" and "viewing (A1)" to understand the design brain of provide preliminary ideas. Both of these two conditions involve the cognitive brain, which includes visual imagery to transport oneself into the landscape, but the "preliminary ideas" condition also engages the brain in generating ideas. Second, the brain region associated with the difference between ""refinement" (B2)" and "copy drawing (A2)" was identified to isolate the graphic design thinking activity. Both of these two conditions engage the brain in drawing a landscape, but the "graphic design thinking" condition also involves the brain in the refinement of the design.

## Results

### Sanity check

To ensure a confident answer to the question of which brain area performs graphic design thinking, we first conducted a sanity check. First, we checked each subject's translational and rotational motion during the functional scans. Both translational and rotational motion not exceed 5mm and 5 degrees was acceptable. The overall motion in each run showed in Table 1. Second, as the experiments included two distinct conditions (with and without hand movements), we analyzed the differences between them to assess whether the primary motor cortex was activated [21].

**Copy drawing (A2) versus viewing (A1).**   We observed "copy drawing (A2)" versus "viewing (A1)" at a threshold of $p < 0.05$ FEW-corrected, extent threshold k > 20 voxels (see Table 2 and Fig 5). This result indicates that the main area of brain activity was the left precentral gyrus (Fig 6), which controls primary motion. Additionally, left-brain activation responded to right-hand movement.

**Refinement (B2) versus preliminary ideas (B1).**   The result of "refinement (B2)" versus "preliminary ideas (B1)" at a threshold of $p < 0.05$ FEW-corrected and extent threshold of

**Table 1. Average framewise displacement during the functional scans.**

| | Translation (mm) | | | Rotation (degres) | | |
|---|---|---|---|---|---|---|
| | x | y | z | pitch | roll | yaw |
| Run1 | 0.10±0.14 | 0.15±0.17 | 0.35±0.27 | 0.01±0.01 | 0.00±0.00 | 0.00±0.00 |
| Run2 | 0.06±0.03 | 0.11±0.13 | 0.27±0.25 | 0.00±0.00 | 0.00±0.00 | 0.00±0.00 |

**Table 2. Brain activation for copy drawing (A2) versus viewing (A1).**

| Brain region | | MNI-coordinates | | | t | P_FWE | Number of voxels |
|---|---|---|---|---|---|---|---|
| | | x | y | z | | | |
| (L) | Precentral gyrus | -32 | -18 | 50 | 15.40 | 0.000 | 5100 |
| (R) | Cerebellum anterior lobe | 14 | -52 | -16 | 14.72 | 0.000 | 2393 |
| (L) | Inferior frontal gyrus | -56 | 8 | 24 | 12.10 | 0.000 | 463 |
| (R) | Inferior frontal gyrus | 56 | 8 | 26 | 10.63 | 0.000 | 306 |
| (R) | Superior parietal lobule (BA7) | 18 | -56 | 60 | 10.29 | 0.000 | 1430 |
| (R) | Frontal lobe sub-gyrus | 24 | -2 | 54 | 9.64 | 0.000 | 238 |
| (L) | Precentral gyrus (BA6) | -48 | -4 | 6 | 7.38 | 0.003 | 25 |
| (L) | Inferior temporal gyrus (BA37) | -48 | -68 | -2 | 7.13 | 0.006 | 98 |
| (R) | Middle temporal gyrus (BA37) | 52 | -58 | -6 | 6.80 | 0.010 | 50 |

k > 20 voxels (Table 3 and Fig 7) showed the left precentral gyrus as the main area of brain activation. This area controls primary motion and is consistent with the expected results for right-hand movement (Fig 8).

The sanity check showed that when participants saw the same stimulus with and without hand movement, there was a difference in the activation of the primary motor cortex. As this confirms that the participants followed the instructions, the collected data could be utilized for the indicated purposes.

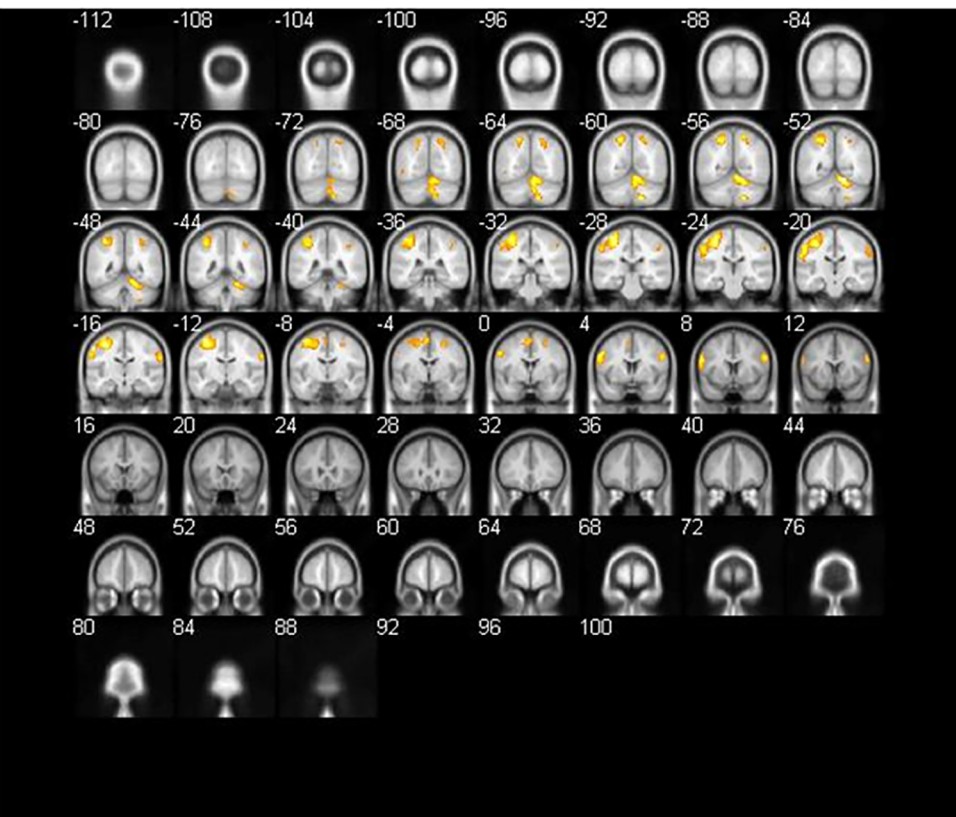

**Fig 5. Brain activation for copy drawing (A2) versus viewing (A1) (coronal section).**

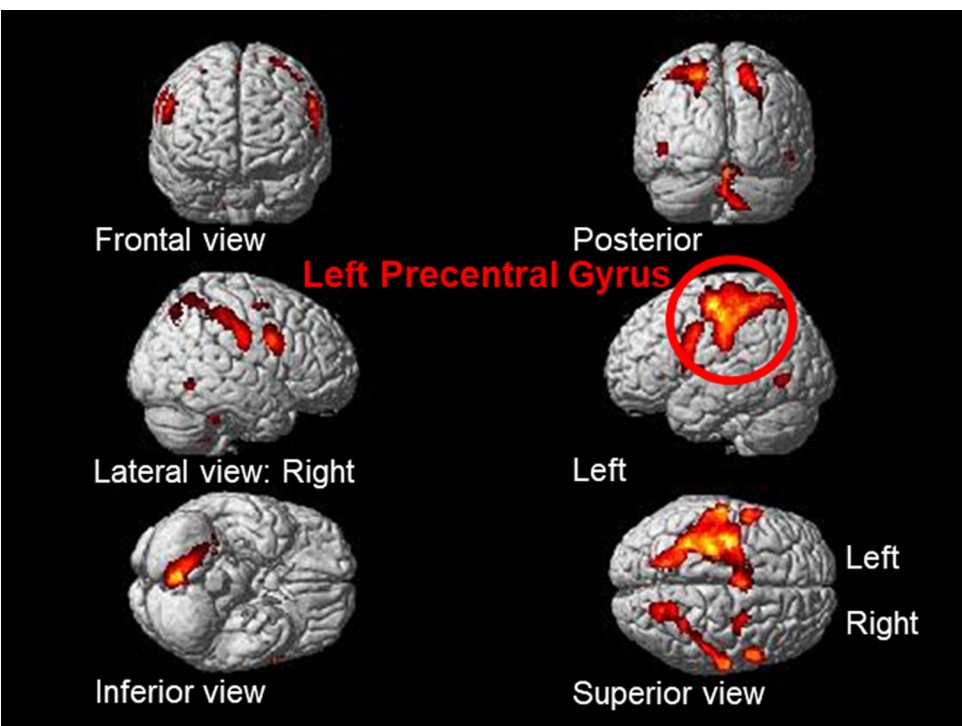

**Fig 6. Brain activation for copy drawing (A2) versus viewing (A1).**

## Hypothesis testing of the design process

The study hypothesis was designed to test the relationship between design activity and the prefrontal cortex. While the preliminary ideas (idea generation) phase activated the right PFC, the refinement (idea production) phase activated the left PFC.

**Preliminary ideas (B1) versus viewing (A1).** We did not find differences in BOLD signals between the "design without drawing (B1)" and "viewing (A1)" conditions at the threshold $p < 0.05$ FEW-corrected and extent threshold $k > 20$ voxels.

**Refinement (B2) versus copy drawing (A2).** For "refinement" versus "copy drawing," activation was observed in the left middle frontal gyrus (peak x, y, z = -52, 20, 30; t = 8.28), which formed part of the DPFC functional region (Table 4 and Fig 9). As this brain region is in charge of cognitive processes, including working memory, cognitive flexibility, and

**Table 3. Brain activation for refinement (B2) versus preliminary ideas (B1).**

| Brain region | | MNI-coordinates | | | t | $P_{FWE}$ | Number of voxels |
|---|---|---|---|---|---|---|---|
| | | x | y | z | | | |
| (R) | Cerebellum anterior lobe | 6 | -66 | -16 | 15.65 | 0.000 | 4443 |
| (L) | Precentral gyrus | -30 | -24 | 58 | 15.54 | 0.000 | 8190 |
| (R) | Inferior frontal gyrus | 56 | 8 | 26 | 14.69 | 0.000 | 653 |
| (R) | Precuneus | 16 | -56 | 58 | 12.57 | 0.000 | 1723 |
| (R) | Middle occipital gyrus (BA19) | 50 | -58 | -8 | 9.46 | 0.000 | 682 |
| (L) | Middle occipital gyrus (BA37) | -44 | -70 | 0 | 8.76 | 0.000 | 245 |
| (L) | Inferior occipital gyrus | -38 | -86 | -6 | 8.47 | 0.001 | 70 |
| (L) | Thalamus | -12 | -18 | 8 | 6.69 | 0.013 | 22 |

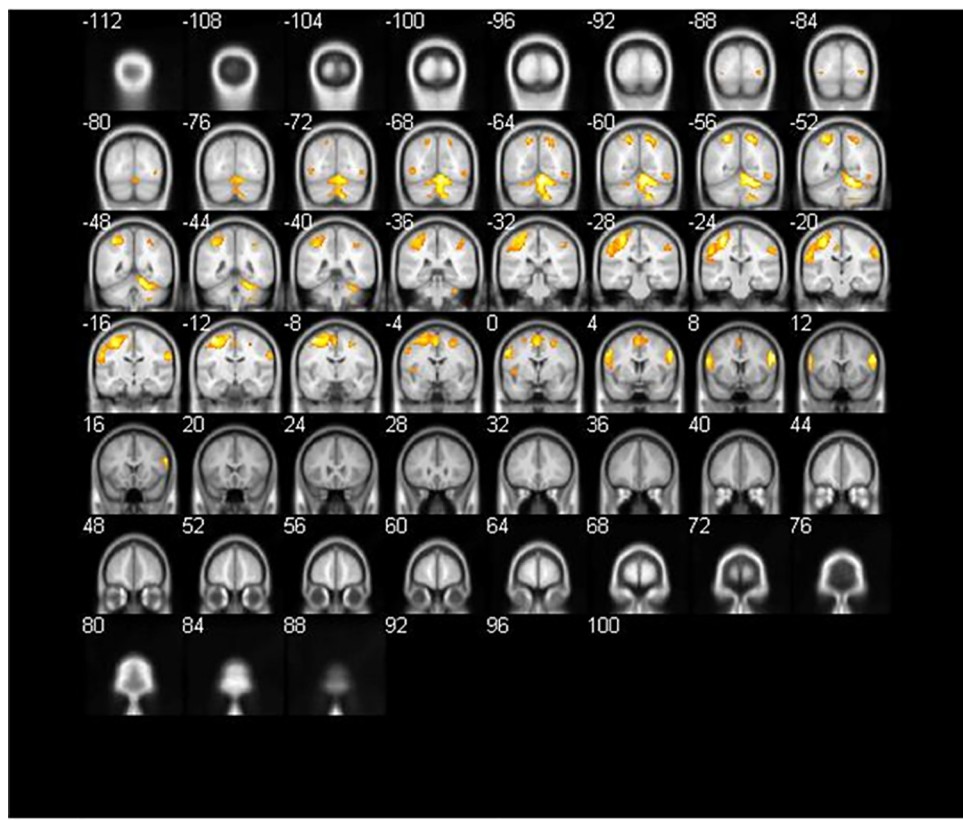

**Fig 7. Brain activation for refinement (B2) versus preliminary ideas (B1) (coronal section).**

planning [22], there is support for the hypothesis that the left PFC is responsible for refinement (idea production).

## The relationship between brain activation parameters and refinement in graphic design thinking scores

**Refinement in graphic design thinking evaluation.**   Inter-judge reliability was quite consistent among the 10 senior experts from the Council of Landscape Architecture Association. They awarded a mean score of 4.60 (SD = 0.75) for refinement in graphic design thinking (Cronbach's alpha = 0.895).

**Correlation of the level of brain activation parameters and refinement in graphic design thinking scores.**   We compared the designers' work, specifically at the level of refinement in graphic design thinking, with the brain activity results. The scores correlated significantly with the Beta value (BOLD-magnitude of brain activation) of "refinement versus copy drawing" in the left middle frontal gyrus ($r = 0.473$, $p < 0.05$) (Fig 10).

## Discussion and conclusions

Based on these findings, the left middle frontal gyrus, which forms part of the PFC, contributed to graphic design thinking in the refinement steps of landscape architecture design (idea production). During the landscape architecture design process, the designers drew and thought simultaneously. The use of paper and pen, as in the real world, helped designers engage in

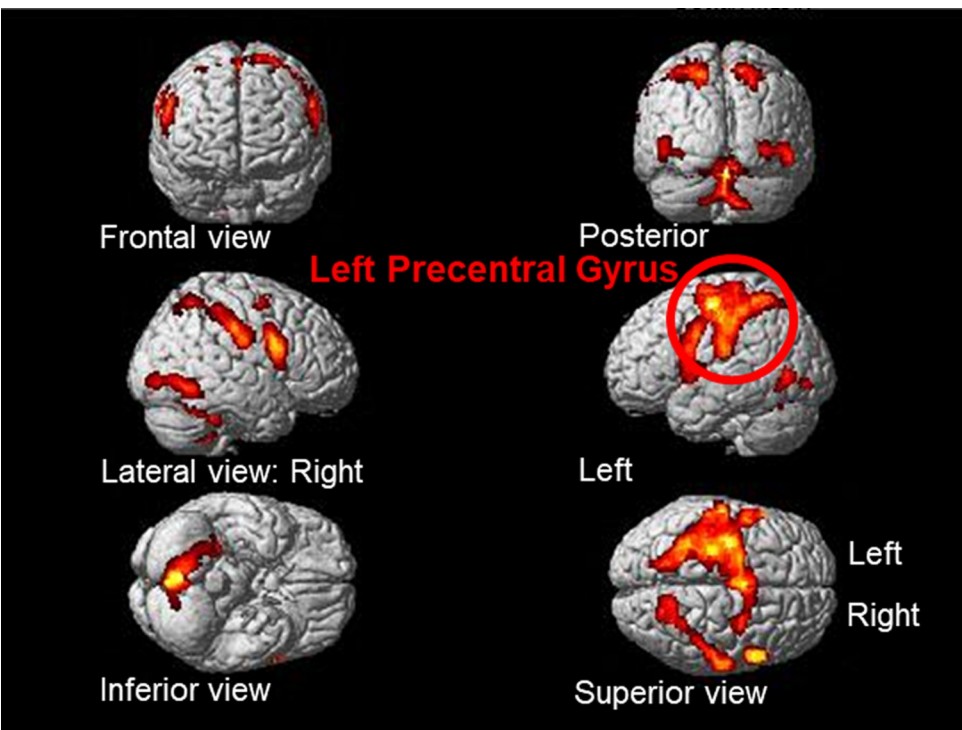

**Fig 8. Brain activation for refinement (B2) versus preliminary ideas (B1).**

graphic design thinking, which may explain the confirmation of the hypothesis that the PFC was active during idea production.

Our results failed to confirm that right PFC activity is associated with the preliminary ideas (idea generation) phase. Having controlled for the imaginary landscape by analyzing "preliminary ideas versus viewing" (both of which are needed to imagine environmental space but differ in relation to graphic design thinking), there was no brain region activity shown for this phase in this study. It is possible that the imaginary landscape could not evoke sufficient graphic design thinking that can be captured in the brain result.

The left and right PFCs play different roles in the design process. Goel [4] proposed the FLLH to explain the relationship between the design process and brain activity. We found support for the hypothesis that the left PFC is involved in the refinement phases. Furthermore, the correlation analysis of refinement in graphic design thinking scores and the contrast between "refinement in graphic design thinking versus copy drawing" demonstrated a positive association with activation of the left middle frontal gyrus, which is part of the left PFC. Higher scores were related to higher activation in the left PFC. Several studies have used methods such as transcranial direct current stimulation (tDCS) to improve creative performance [23], suggesting a new direction for further study involving stimulation of the left middle prefrontal brain area to improve design skills.

**Table 4. Activation peaks for refinement (B2) versus copy drawing (A2).**

| Brain region | | MNI-coordinates | | | t | P_FWE | Number of voxels |
|---|---|---|---|---|---|---|---|
| | | x | y | z | | | |
| (L) | Middle Frontal Gyrus | -52 | 20 | 30 | 8.28 | 0.001 | 100 |

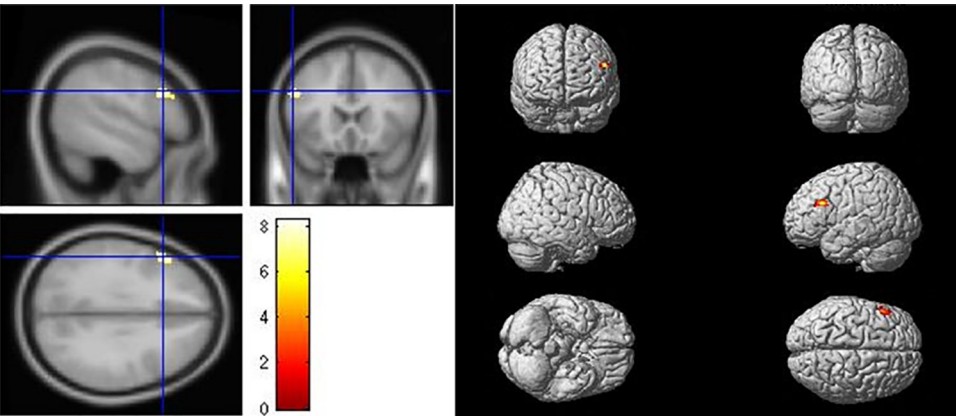

**Fig 9. Brain activation for refinement (B2) versus copy drawing (A2).**

The present results confirm that the left middle frontal gyrus is involved in graphic design thinking in landscape architecture design and can be linked to the idea production phase of creativity [4, 12, 13, 15, 16]. Besides graphic design thinking, these findings also relate to other elements, such as emotions and personal experiences [7, 8, 12, 15]. The landscape architecture design process, as it unfolds in the designer's brain, warrants further experiment-based research to address a number of questions. First, how does emotion impact the designer while engaged in landscape architecture design, and how is the brain activated during different phases of the design experience? Second, what do differences between experts and non-experts reveal about the characteristics of the designer's brain? There is also more to be learned about the mechanisms of creativity and methods for training our brains to be more innovative. Further research could consider other experimental tasks or repeat more tasks to maximize evoked changes in brain activation. Moreover, limited to the experimental tasks printed and bound on paper, the task sequence in this study has only two counterbalancing versions. It is better to randomize the experimental tasks to avoid ordering effect. These preliminary pilot

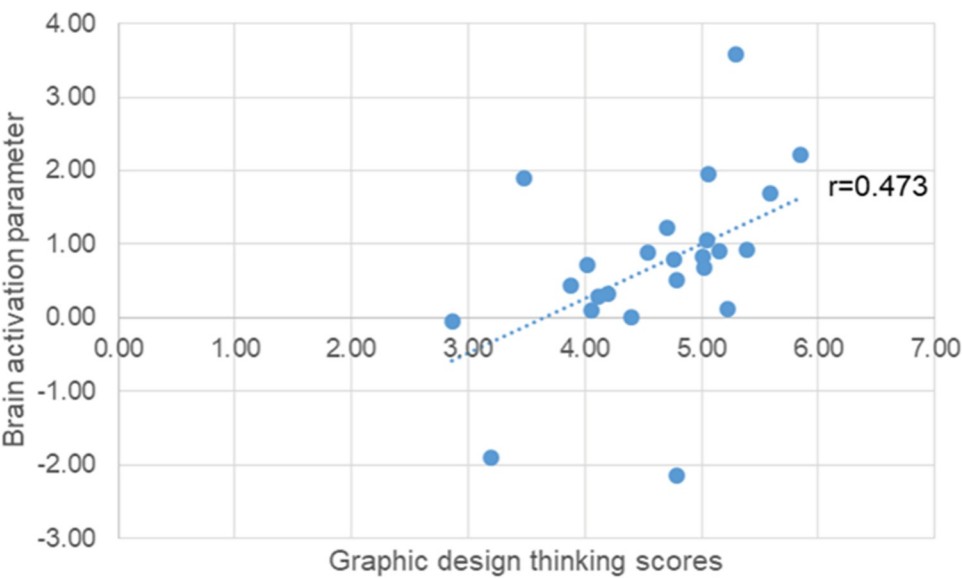

**Fig 10. Correlation of brain activation parameters and refinement in graphic design thinking scores.**

experiment results will be further analyzed, developed, and replicated in pursuing this psychological line of investigation into the links between the process of landscape architecture design and human brain activity.

## Acknowledgments

We are grateful to all the participants for giving their time to the experiment. Special thanks to the team members of the Imaging Center for Integrated Body, Mind and Culture Research for assisting with the fMRI study, and Dr. Vinod Goel for offering suggestions on an earlier version of this paper. This journal article has been modified from unpublished master's thesis on *Neural Correlates of Landscape Design Creativity*: *An fMRI Study*, Department of Horticulture and Landscape, National Taiwan University.

## Author Contributions

**Conceptualization:** Chun-Yen Chang.

**Formal analysis:** Yu-Ping Tsai, Tsung-Ren Huang.

**Funding acquisition:** Chun-Yen Chang.

**Investigation:** Yu-Ping Tsai.

**Methodology:** Tsung-Ren Huang, William C. Sullivan, Chun-Yen Chang.

**Writing – original draft:** Yu-Ping Tsai.

**Writing – review & editing:** Yu-Ping Tsai, Shih-Han Hung, William C. Sullivan, Shih-An Tang, Chun-Yen Chang.

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
