## [Decision Letter · Decision Letter 0]

27 Oct 2020

PONE-D-20-09407

What part of the brain is involved in graphic thinking for landscape design?

PLOS ONE

Dear Dr. Chang,

Thank you for submitting your manuscript to PLOS ONE. After careful consideration, we feel that it has merit but does not fully meet PLOS ONE’s publication criteria as it currently stands. Therefore, we invite you to submit a revised version of the manuscript that addresses the points raised during the review process.

Please pay particular attention to addressing Reviewer 1's concerns regarding the definition of key terms and explanation of the methodology. Providing these clarifications will be essential for a more thorough examination of the results and their interpretation.

We look forward to receiving your revised manuscript.

Kind regards,

Jamie Males

Senior Editor

PLOS ONE

Journal Requirements:

2. We note that Figure 3 in your submission contains copyrighted images.

All PLOS content is published under the Creative Commons Attribution License (CC BY 4.0), which means that the manuscript, images, and Supporting Information files will be freely available online, and any third party is permitted to access, download, copy, distribute, and use these materials in any way, even commercially, with proper attribution. For more information, see our copyright guidelines: http://journals.plos.org/plosone/s/licenses-and-copyright.

We require you to either (1) present written permission from the copyright holder to publish this figure specifically under the CC BY 4.0 license, or (2) remove the figures from your submission:

a. You may seek permission from the original copyright holder of Figure 3 to publish the content specifically under the CC BY 4.0 license.

b.    If you are unable to obtain permission from the original copyright holder to publish this figure under the CC BY 4.0 license or if the copyright holder’s requirements are incompatible with the CC BY 4.0 license, please either i) remove the figure or ii) supply a replacement figure that complies with the CC BY 4.0 license. Please check copyright information on all replacement figures and update the figure caption with source information. If applicable, please specify in the figure caption text when a figure is similar but not identical to the original image and is therefore for illustrative purposes only.

3. Thank you for including your ethics statement:

"Each participant gave their written informed consent as approved by the Research Ethics Committee, National Taiwan University [approval number 201411HM024]."

a) Please amend your current ethics statement to confirm that your named institutional review board or ethics committee specifically approved this study.

Reviewers' comments:

Reviewer's Responses to Questions

**Comments to the Author**

1. Is the manuscript technically sound, and do the data support the conclusions?

Reviewer #1: No

Reviewer #2: Yes

2. Has the statistical analysis been performed appropriately and rigorously? 

Reviewer #1: No

Reviewer #2: Yes

3. Have the authors made all data underlying the findings in their manuscript fully available?

Reviewer #1: No

Reviewer #2: Yes

4. Is the manuscript presented in an intelligible fashion and written in standard English?

Reviewer #1: Yes

Reviewer #2: Yes

5. Review Comments to the Author

Reviewer #1: The manuscript describes an fMRI study of 24 designers engaging in various aspects of the landscape design task. It is concluded that "designs involving more graphic thinking exhibit significantly more activity in left prefrontal cortex."

This is a potentially interesting manuscript but is not currently ready for publication.

Issues that need to be addressed:

The introduction is overly broad and uninformative. The authors start by asking "what is creativity?" But do not provide an answer. They cannot provide an answer because there are no good answers to the question. So why ask the question?

2. They introduce the notions of "thoughtful creativity" and "spontaneous creativity" without explaining them. I do not know what these terms refer to. They then refer to some findings regarding creativity, but it is not clear what type of creativity they are referring to.

3. More generally, the manuscript is about "graphic thinking"… So why start talking about creativity? The discussion is essentially vacuous and unnecessary.

4. Their references regarding problem-solving are dated.

5. I do not understand what they did with the Torrance test.

These issues can be easily dealt with by simply rewriting the introduction and reducing the scope of the manuscript. However, there are some methodological shortcomings that do not have an easy solution.

In particular, what is graphic thinking? What cognitive activities are actually being measured?

On the one hand the authors are talking about problem solving in terms of problems scoping, preliminary solutions, refinement, and detailing (figure 1). On the other hand, their task involves viewing, copying drawing, brainstorming, graphic thinking (figure 4). How does the one vocabulary map onto the other? Even more importantly, what is happening during the 60 seconds allowed for each phase? What are the participants doing? What cognitive processes are being engaged? What behavioural data can the authors provide to support their conjecture? There is no way of knowing based on the information provided. If we cannot answer these questions, we cannot even begin to interpret the results.

There are several other methodological shortcomings I could point out. However, I will stop here because this is a deal breaker. If the authors cannot address this crucial issue, there are no sound results to publish.

Reviewer #2: This paper addressed a very important question about the brain's underpinning of human behavior, specifically creativity and graphical design. The methodology and results are convincing, and the discussion is to the point. As far as this Reviewer is concerned this version can be published as is.

6. PLOS authors have the option to publish the peer review history of their article (what does this mean?). If published, this will include your full peer review and any attached files.

Reviewer #1: No

Reviewer #2: No

---

## [Author Response · Author response to Decision Letter 0]

10 Dec 2020

Thank you for inviting us to submit a revised version of the manuscript entitled, “What part of the brain is involved in graphic thinking for landscape design? [PONE-D-20-09407]” to PLOS ONE. We really appreciate the time and effort you and each of the reviewers have dedicated to providing insightful feedback on ways to strengthen our paper. 

It is with great pleasure that we resubmit our article for further consideration. We have included a point-by-point response to the questions and comments delivered in your letter dated October 27, 2020. The revised words and sentences are highlighted in gray.

We hope that the revised manuscript is clearer in definition of key terms of the graphic thinking and design process, and in the more explanation of the methodology. We hope these revisions satisfactorily address all the issues and concerns you and the reviewers have noted.

---

## [Decision Letter · Decision Letter 1]

24 Feb 2021

PONE-D-20-09407R1

What part of the brain is involved in graphic thinking for landscape design?

PLOS ONE

Dear Dr. Chang,

Thank you for submitting your manuscript to PLOS ONE. After careful consideration, we feel that it has merit but does not fully meet PLOS ONE’s publication criteria as it currently stands. Therefore, we invite you to submit a revised version of the manuscript that addresses the points raised during the review process.

The manuscript has been re-evaluated by two reviewers, and their comments are available below. You will see Reviewer 2 has commented in the improvement of the manuscript. However, Reviewer 3 has raised critical concerns and the manuscript will need significant revision before it can be considered for publication – you should anticipate that the reviewers will be re-invited to assess the revised manuscript, so please ensure that your revision is thorough. I have outlined some of the key concerns noted by the reviewers below, but you should respond to all concerns mentioned by the reviewers in your response-to-reviewers document. 

The key concern noted by Reviewer 3 relate to the definition of terms, the need for additional supporting references, and clarification regarding the fMRI analysis of the ventricles. These issues impact the interpretation of the results and should be explored.

We look forward to receiving your revised manuscript.

Kind regards,

Danielle Poole

Staff Editor

PLOS ONE

Reviewers' comments:

Reviewer's Responses to Questions

**Comments to the Author**

1. If the authors have adequately addressed your comments raised in a previous round of review and you feel that this manuscript is now acceptable for publication, you may indicate that here to bypass the “Comments to the Author” section, enter your conflict of interest statement in the “Confidential to Editor” section, and submit your "Accept" recommendation.

Reviewer #2: All comments have been addressed

Reviewer #3: (No Response)

2. Is the manuscript technically sound, and do the data support the conclusions?

Reviewer #2: Yes

Reviewer #3: No

3. Has the statistical analysis been performed appropriately and rigorously? 

Reviewer #2: Yes

Reviewer #3: I Don't Know

4. Have the authors made all data underlying the findings in their manuscript fully available?

Reviewer #2: Yes

Reviewer #3: No

5. Is the manuscript presented in an intelligible fashion and written in standard English?

Reviewer #2: Yes

Reviewer #3: Yes

6. Review Comments to the Author

Reviewer #2: The authors have addressed the use of the term creativity, which made the revised paper more scholarly than the original version.

Reviewer #3: This paper analyses the brain areas activated in a design task where designers were asked to perform a landscape design task under the fMRI scanner in two stages: brainstorming without drawing and graphic thinking where they draw and refine the ideas they generated during the brainstorming stage. There were two control conditions were of viewing a plan and imagining being there and another was to copy a plan. I have reservations about the design of the study and the depth of conceptual underpinnings of the study. Most of the terms are loosely defined and leads to confusion. The introduction and discussion are not detailed and severely lack references.

1. The title of the paper, “What part of the brain is involved in graphic thinking for landscape design?” is misleading. This insinuates a specific part of the brain is involved in graphic thinking, specifically for landscape design. This is not true for many reasons, but the most pertinent is that the authors have not checked any other graphic thinking in any other creative fields, like book cover design or jewelry design or product design. Some comments on the commonality and differences between the other kinds of graphic thinking would have put this study in perspective.

2. The first line of the abstract is problematic – “Graphic thinking is a key skill for landscape designers, but little is known about the links between design thinking and brain activity.” How are graphic thinking and design thinking related? There is a large body of work done on design thinking, but none of them are cited (later in the article), nor are the connection between the theories in the design creativity and graphic thinking for landscape design explained.

3. The introduction lacks scholarly depth and references.

4. It is not clear what the authors describe as “graphic thinking.” The definition, without elaboration; “thinking assisted by sketching” is not adequate. It also implies that sketching and thinking are separate, and that sketching can occur without thinking? Although it might be true in common parlance, sketching “without thinking” also is a complex cognitive task. Also, how does graphic thinking differ from visual imagery, visual creativity etc.

5.There are no references provided for the argument that “The landscape design process contains four steps” lines 37-46. This is vital.

6. There are no references for “In real-world problem-solving, the left and right prefrontal cortices (PFC) are responsible for different functions.” Line 49

7. “The landscape designer develops a concept through brainstorming and then processes ideas, using graphic thinking to refine the design” Line 54 – There are too many terms here that are not well defined – brainstorming? What does that mean? How is it different from “processes ideas”?

8. “Although Goel advanced this hypothesis to explain the neural mechanisms underpinning the design process, there was no empirical evidence to validate this assumption.” Line 61. This is not true, Goel has published lesion-studies (including an architect with a lesion) and brain imaging studies to support his theory.

Procedure

9. “brainstorming, using geometric illustrations to envision a landscape design (idea generation) (B-1); and graphic thinking, using drawing to modify the design from the brainstorming session (idea production) (B-2)” line 122 - We have no idea what the participants were doing in the brainstorming session. So, it is wrong to say that the designers are modifying the design from the brainstorming session. They might be producing the ideas and drawings together, as claimed by the definition of graphic thinking “thinking assisted by sketching”

10. “In the “viewing” condition, participants were asked to imagine they were in the

environment that the plan presented.” - Line 126 – This is not a simple viewing condition, but a complex cognitive task of transporting oneself into an imaginary place. This aspect has not been interpreted correctly in the results nor explained in the discussion.

11. “To avoid any ordering effect, the two tasks were run as ABAB BABA or BABA ABAB” - line 137. This kind of design does not remove the ordering effect as A always follows B, only the starting condition has been changed. To truly remove the ordering effect, the presentation must have been randomized.

Results

12. “The results show brain activation at the lateral ventricle, which contains cerebrospinal

fluid that protects the brain from impact injury and assists in nutrient cycling and waste

removal. It is not responsible for specific cognitive function. On that basis, our hypothesis that the preliminary solution (idea generation) phase activates the right PFC was not verified” – 254. I am confused by this result. How can hemodynamic activity be seen in the ventricles? I am not an expert in fMRI analysis, but this is very suspect.

13. “As this brain region is in charge of cognitive processes, including working memory, cognitive flexibility, and planning [10], there is support for the hypothesis that the left PFC is responsible for refining and detailing (idea production)” - This contradicts authors claim in lines 37 to 40. Idea production happens before refining and detailing stage.

Discussion

14. “the only region activated was the lateral ventricle. Further experiments are needed to facilitate interpretation of this finding” 302 – The ventricles cannot show hemodynamic activity.

15. Besides graphic thinking, this also relates to other elements, such as emotions, memories, and

experiences.” Line 315 - please cite references

7. PLOS authors have the option to publish the peer review history of their article (what does this mean?). If published, this will include your full peer review and any attached files.

Reviewer #2: No

Reviewer #3: No

---

## [Author Response · Author response to Decision Letter 1]

9 Apr 2021

Thank you for inviting us to submit a revised version of our manuscript entitled, “What part of the brain is involved in graphic design thinking in landscape architecture? [PONE-D-20-09407R1]” to PLOS ONE. We really appreciate the time and effort you and each of the reviewers have dedicated to providing insightful feedback on ways to strengthen our paper.

It is with great pleasure that we resubmit our article for further consideration. We have included a point-by-point response to the questions and comments delivered in your letter dated February 24, 2021. The revised words and sentences are highlighted in gray.

We hope that the revised manuscript is clearer in the definition of terms, the need for additional supporting references, and clarification regarding the fMRI analysis of the ventricles. We hope these revisions satisfactorily address all the issues and concerns you and the reviewers have noted. Thank you.

---

## [Decision Letter · Decision Letter 2]

10 Aug 2021

PONE-D-20-09407R2

What part of the brain is involved in graphic design thinking in landscape architecture?

PLOS ONE

Dear Dr. Chang,

Thank you for submitting your manuscript to PLOS ONE. After careful consideration, we feel that it has merit but does not fully meet PLOS ONE’s publication criteria as it currently stands. Therefore, we invite you to submit a revised version of the manuscript that addresses the points raised during the review process.

I personally assessed your manuscript. I acknowledge the improvement introduced during the various revisions, and the rather cumbersome revision history. I appreciate also the overall interest of the topic.

At this point, in fairness to authors, I just ask for these minor changes:

Line 217: use proper capitalization, 3 T or 3 tesla (Tesla => tesla)

Line 218 please double check T1 scan parameters, these does not look parameters suitable for T1 weighting. I’ll check the images on the repository (see below)

Line 221 TE= 30 ms (add "s")

Line 222 matrix size=64x64x45, without mm3 unit (these are pure numbers)

Line 226 please delete "According to the preprocessing steps in the SPM8 manual" remark. It sounds naive.

I must highlight that the MRI protocol is remarkably old style. It is very odd, especially considering the high performance scanner that was used. Is there any reason behind this methodological choice? You could add a remark like "no acceleration was used to minimize the scans sensitivity to motion", that on my eyes is still unsatisfactory, but at least hints why a 15 yrs old experimental approach was used

Sanity check section: please add information on the overall motion of the subjects during the functional scans, eg a table with average framewise displacement in each run

Line 350: what is the "preprint booklet"? It is the first time this term appears. Please use the same wording used in methods.

Data availability statement is missing in the manuscript, and the version in front matter is misleading ("All relevant data are within the manuscript.", but no raw data is actually available in the manuscript). MRI scans and scores from each subject must be made available on a public repository before acceptance, and the link must be included in the final manuscript version.

We look forward to receiving your revised manuscript.

Kind regards,

Federico Giove, PhD

Academic Editor

PLOS ONE

Journal Requirements:

Reviewers' comments:

Reviewer's Responses to Questions

**Comments to the Author**

1. If the authors have adequately addressed your comments raised in a previous round of review and you feel that this manuscript is now acceptable for publication, you may indicate that here to bypass the “Comments to the Author” section, enter your conflict of interest statement in the “Confidential to Editor” section, and submit your "Accept" recommendation.

Reviewer #2: All comments have been addressed

2. Is the manuscript technically sound, and do the data support the conclusions?

Reviewer #2: Yes

3. Has the statistical analysis been performed appropriately and rigorously? 

Reviewer #2: Yes

4. Have the authors made all data underlying the findings in their manuscript fully available?

Reviewer #2: Yes

5. Is the manuscript presented in an intelligible fashion and written in standard English?

Reviewer #2: Yes

6. Review Comments to the Author

Reviewer #2: The authors have addressed reviewers' criticisms, comments, and suggestions adequately, and this version of the paper is good now. I recommend acceptance for publication.

7. PLOS authors have the option to publish the peer review history of their article (what does this mean?). If published, this will include your full peer review and any attached files.

Reviewer #2: No

---

## [Author Response · Author response to Decision Letter 2]

21 Sep 2021

Thank you for inviting us to submit a revised version of our manuscript entitled, “What part of the brain is involved in graphic design thinking in landscape architecture? [PONE-D-20-09407R2]” to PLOS ONE. We really appreciate the time and effort you and each of the reviewers have dedicated to providing insightful feedback on ways to strengthen our paper.

It is with great pleasure that we resubmit our article for further consideration. We have included a point-by-point response to the questions and comments delivered in your letter dated August 10, 2021. The revised words and sentences are highlighted in gray. We revised the manuscript according to your suggestions and upload the raw dataset on a public repository (https://doi.org/10.7910/DVN/ZHZG5C). We hope these revisions satisfactorily address all the issues and concerns you have noted.

Thank you again for your consideration of our revised manuscript.

---

## [Editor Report · Decision Letter 3]

28 Sep 2021

What part of the brain is involved in graphic design thinking in landscape architecture?

PONE-D-20-09407R3

Dear Dr. Chang,

We’re pleased to inform you that your manuscript has been judged scientifically suitable for publication and will be formally accepted for publication once it meets all outstanding technical requirements.

Kind regards,

Federico Giove, PhD

Academic Editor

PLOS ONE
---

## [Editor Report · Acceptance letter]

15 Dec 2021

PONE-D-20-09407R3 

What part of the brain is involved in graphic design thinking in landscape architecture? 

Dear Dr. Chang:

I'm pleased to inform you that your manuscript has been deemed suitable for publication in PLOS ONE. Congratulations! Your manuscript is now with our production department. 

Kind regards, 

on behalf of

Dr. Federico Giove 

Academic Editor

PLOS ONE